# Radiotherapy and Immunotherapy in Lung Cancer

**DOI:** 10.3390/biomedicines11061642

**Published:** 2023-06-06

**Authors:** Kristin Hsieh, Daniel R. Dickstein, Juliana Runnels, Eric J. Lehrer, Kenneth Rosenzweig, Fred R. Hirsch, Robert M. Samstein

**Affiliations:** 1Department of Radiation Oncology, Icahn School of Medicine at Mount Sinai, New York, NY 10029, USA; 2Center for Thoracic Oncology, Tisch Cancer Institute, Icahn School of Medicine at Mount Sinai, New York, NY 10029, USA

**Keywords:** radiotherapy, immunotherapy, lung cancer, NSCLC, SCLC

## Abstract

The emergence of immune checkpoint inhibitors (ICIs) as a pillar of cancer treatment has emphasized the immune system’s integral role in tumor control and progression through cancer immune surveillance. ICIs are being investigated and incorporated into the treatment paradigm for lung cancers across stages and histology. To date, definitive concurrent chemoradiotherapy followed by consolidative durvalumab is the only National Comprehensive Cancer Network’s recommended treatment paradigm including radiotherapy with ICI in lung cancers, although there are other recommendations for ICI with chemotherapy and/or surgery. This narrative review provides an overall view of the evolving integration and synergistic role of immunotherapy and radiotherapy and outlines the use of immunotherapy with radiotherapy for the management of small cell lung cancer and non-small cell lung cancer. It also reviews selected, practice-changing clinical trials that led to the current standard of care for lung cancers.

## 1. Introduction

Despite medical advances, lung cancer is the second most commonly diagnosed cancer and the leading cause of cancer-related death worldwide [1]. It also continues to be the third most common cancer and the leading cause of cancer-related deaths in the United States [2]. The two main types of lung cancer are small cell lung cancer (SCLC) and non-small cell lung cancer (NSCLC), both with similar treatment paradigm that typically includes a combination of focal therapy and systemic therapy depending on the stage of disease.

Decades of research have implicated the crucial role of the immune system in tumor surveillance and control and the potential for immune-targeted therapies. Cancer immune surveillance is the process in which the immune system controls tumor growth and progression [3]. However, tumors may avoid immune surveillance by a variety of immunosuppressive mechanisms now recognized as a critical hallmark of cancer [4]. These mechanisms, including inducing upregulation of immune inhibitory pathways or checkpoints as well as induction of an immune suppressive tumor microenvironment, may also predict a patient’s prognosis and are observed in patients with SCLC and NSCLC [3,5,6,7].

With the introduction of immunotherapy (IO) as a key systemic agent in both types of lung cancer, this article outlines the integration of IO with radiotherapy (RT) for the management of SCLC and NSCLC. Firstly, it provides an overall view of the evolving synergistic role of IO and RT, utilizing the paradigm of adding IO to RT in non-metastatic disease to decrease the risk of local and distant failure versus adding RT to IO in metastatic disease to enhance immunotherapeutic efficacy. Given the rapid integration of IO to the treatment of lung cancers, the synergistic role of RT has been tested in prospective trials across disease types and stages and will be reviewed. This narrative review focuses on the most commonly utilized and currently FDA approved IO agents, immune checkpoint inhibitors including antibodies targeting CTLA4 and PD-1/PD-L1. It also reviews selected, practice-changing clinical trials that integrate IO and RT.

## 2. Rationale of Integrating IO and RT

The critical role of RT in the management of lung cancers is evolving due to the role of novel therapeutic approaches [8,9,10,11]. Despite the exciting potential to integrate IO with RT given their biologic basis, few articles have focused on the synergy between IO and RT in lung cancer, with most focusing only on NSCLC [12,13,14].

There are multiple rationales for integrating IO and RT. Firstly, RT targets cancer cells within the treatment field by causing direct and indirect DNA damage. However, its effect may not only be limited to the treatment field. Prior studies and case reports across disease sites have noted tumor control outside the RT treatment field in a phenomenon known as an abscopal effect [15,16]. While the dramatic abscopal effects have ultimately proven a rare phenomenon to date, it emphasizes the immunogenic aspects of RT and its potential synergy with immune-directed treatments. Research have elucidated that RT can modulate tumor immunogenicity and has previously been reviewed elsewhere [15,16]. Specifically, RT damages cancer cells and thereby exposes tumor-specific antigens, particularly damage-associated molecular patterns (DAMPs) such as ATP and HMGB1 [17]. These DAMPs can interact with receptors on dendritic cells, leading to their stimulation, maturation, and activation, which can then lead to the recruitment, activation, and proliferation of natural killer cells and CD4+ and CD8+ T cells [18,19,20]. At the same time, RT also activates inflammatory pathways through a variety of pathways including immunogenic cell death and antiviral interferon-response genes. Taken together, these features have suggested that local irradiation may serve as a tumor in situ vaccine providing high doses of both antigen and adjuvant inflammatory signals to boost or re-invigorate anti-tumor immune responses.

Given IO promotes the immune system to recognize and target cancer cells, IO and RT together may have synergistic effect temporally and spatially [21,22,23]. Time to clinical and/or radiographic responses to IO may take weeks to month and hyperprogression of disease in approximately 4–29% of patients receiving IO have been reported [24,25]. Thus, RT may complement IO as a bridging strategy for disease control [26]. Responders to IO may also be continuously treated for an undefined period of time, thereby increasing the possibility of physical and financial burden for patients [27]. Consequently, additional IO clinical trials that not only focus on optimizing duration of IO and potential early IO cessation but also explore the role of adding IO to maximize durable response are crucial.

There are two major paradigms for integrating IO and RT. In the setting of non-metastatic disease, adding IO to the standard of care definitive RT may decrease the risk of local and distant failure. The proof of concept has been demonstrated in a phase II study for early stage NSCLC but was rapidly integrated as a new standard of care for locally-advanced NSCLC based on the PACIFIC trial [28,29,30]. In the setting of metastatic disease, adding RT to IO may enhance the systemic efficacy of IO [31]. Two phase I/II clinical trials have explored this concept and found adding RT to IO in metastatic NSCLC are well-tolerated with good disease response and patient outcome [32]. There is also an emerging role of RT for oligometastatic disease with the emerging concept that providing local control in a limited number of metastatic sites may play an important role in systemic disease control and delayed progression. Selected ongoing, multi-institutional SCLC and NSCLC trials sorted using the aforementioned paradigm are listed in tables below and will be reviewed.

Although there is exciting potential in integrating IO and RT, any possible treatment interference and synergistic toxicity must always be considered and assessed. While RT may help activate the immune system, it may also induce lymphopenia and thereby immunosuppression [33]. Thus, further clinical trials should not only assess the role of RT, but also the methods to optimize RT with IO, specifically the effect of the total RT dose and fractionation on the immune response and the timing schedule when integrating IO and RT [34]. Preclinical studies have explored some of the aforementioned aspects. Specifically, for anti-PD-1/PD-L1 therapy, a prior study has found a higher PD-L1 expression in vitro after RT of 10 Gy (5 Gy given 24 h apart) compared to 5 Gy and at timepoint 48 h post-RT compared to 24 h [35]. When the study was repeated in vivo with RT of 10 Gy, a higher PD-L1 expression was again found at 48 h post-RT compared to 24 h. This research group proceeded with concurrent anti-PD-L1 first given between the two 5 Gy fractions and found statistically significant slower growth rate of the irradiated tumor in the murine model when compared to the RT alone group. Interestingly, they found the non-irradiated tumors in the IO with RT group also had a statistically significant slower growth rate than those in the IO alone or RT alone group. Other groups also demonstrated that concurrent IO and RT is beneficial [36,37,38]. There are also several lung cancer-specific studies. Wang et al. have found transient PD-L1 expression in circulating tumor cells during RT in NSCLC patients, but they did not seek to identify the time of peak PD-L1 expression to optimize IO delivery time [39]. However, despite in vitro and in vivo data suggesting the benefits of concurrent RT and IO, some of the large randomized clinical trials are utilizing sequential RT and IO. There is still much to explore with RT and IO to identify the optimal regimen.

Regarding synergistic side effects, pneumonitis is a potential side effect of both IO and RT in lung cancers, though it may have distinct spatial features on imaging depending on the specific cause [40]. A prior study examining three consecutive phase I trials involving stereotactic body radiotherapy (SBRT) and IO demonstrated an excellent local control with an acceptable toxicity profile, specifically a rate of grade 3+ pneumonitis of 8.1% [41]. This study also demonstrated the effectiveness and safety of the established dose-volume lung constraints. Additional studies have demonstrated no statistically significant difference in the rate of pneumonitis when integrating IO and RT, thereby the safety of utilizing both treatment modalities [42,43,44,45]. Additional side effects of RT and IO have previously been reviewed and will not be reviewed in this article [46]. The next sections will provide an overview of management-changing clinical trials and offer insight into the next stage of clinical trials.

## 3. SCLC

### 3.1. Background

Representing approximately 15% of all lung cancers, SCLC is a poorly differentiated, high-grade neuroendocrine tumor that occurs primarily in the central airways [47]. Smoking is the primary risk factor for SCLC, and a decrease in smoking has resulted in a decrease in the incidence of SCLC [47]. Approximately 35,000 cases of SCLC are diagnosed each year with 30% of the cases being limited disease (LS-SCLC), defined as disease localized in one lung, and 70% extensive disease (ES-SCLC), defined as disease spreading to other parts of the body [48]. SCLC is lethal with a median survival of 20 months for LS-SCLC and 10 months for ES-SCLC [49].

### 3.2. Treatment Paradigm for LS-SCLC

The standard for treatment of limited stage LS-SCLC includes chemotherapy, usually cisplatin and etoposide for four cycles, with concurrent RT starting at cycle one or two [50]. Additionally, surgery may be useful in early limited stage SCLC, with adjuvant RT to follow in the setting of certain histopathological features. The regimen for definitive concurrent radiation is 45 Gy over the course of 30 fractions with 2 fractions daily established over two decades per the Intergroup trial 0096 [51]. Daily fractionation to 60–70 Gy in 2 Gy fractions is often offered as per the CONVERT and RTOG 0538 trials, although they were not shown to be superior [52,53]. Additionally, prophylactic cranial irradiation (PCI) to a dose of 25 Gy in 10 fractions for responders after restaging may also be recommended. Of note, there are no existing trials that show PCI confers a survival benefit. There are results from a phase II study exploring concurrent and maintenance durvalumab in LS-SCLC that show clinical efficacy [54]. However, there is no published phase III data supporting use of IO in LS-SCLC, although the results of the ongoing LU005 trial testing the addition of concurrent and maintenance atezolizumab (anti-PD-L1), NCT04624204 trial testing the addition of olaparib with pembrolizumab, and ADRIATIC trial assessing the addition of durvalumab with or without tremelimumab are eagerly anticipated. Ongoing trials are outlined in Table 1.

Still, there has been some research, while unsuccessful, investigating the role of the immune system in the management of LS-SCLC. Bottomley et al. investigated an adjuvant vaccine therapy for patients with LS-SCLC who had a major response to chemoradiotherapy [59,60]. In this randomized trial comparing 5 different vaccinations in 515 patients with LS-SCLC, there was no improvement in overall or progression-free survival. While research is ongoing to investigate the role of the immune system in the treatment of LS-SCLC, there is a more established role of IO to treat ES-SCLC.

### 3.3. Treatment Paradigm for ES-SCLC

The treatment for ES-SCLC includes platinum-based chemotherapy for four cycles with concurrent and then maintenance atezolizumab or durvalumab, with palliative RT to symptomatic sites [50]. Atezolizumab is an antibody immune checkpoint inhibitor that binds to PD-L1 to enhance tumor-specific T-cell immunity. IMpower133 was an international, double-blind, randomized phase III study with 403 treatment-naïve patients from 22 countries that sought to evaluate the combination of checkpoint inhibition with cytotoxic chemotherapy on progression-free and overall survival [61]. The median overall survival was 12.3 months and 10.3 month, while the progression-free survival was 5.2 and 4.3 months for the atezolizumab and placebo groups, respectively. This trial illustrated that the addition of atezolizumab to cytotoxic chemotherapy results in better disease and survival outcomes compared to chemotherapy alone albeit with an incremental improvement leaving room for further advances. Still, it is important to note that there were immune-related adverse events, including rash, hepatitis, pneumonia, and colitis, that were more common in the atezolizumab group compared to the standard cohort similar to those seen in other cancer types with these agents.

The CASPIAN trial is an ongoing phase III trial investigating durvalumab with or without tremelimumab in combination with platinum based therapy (cisplatin or carboplatin) [62,63]. Treatment-naïve patients were randomized to durvalumab plus platinum-etoposide, durvalumab plus tremelimumab plus platinum-etoposide, or platinum-etoposide alone. Patients received four cycles of platinum-etoposide and durvalumab with or without tremelimumab followed by maintenance durvalumab with or without platinum-etoposide every three weeks plus prophylactic cranial irradiation. The primary endpoint was overall survival and a planned interim analysis showed that durvalumab plus platinum-etoposide was associated with significantly improved overall survival with a median survival of 13 months and 10.3 months in the durvalumab plus platinum-etoposide and platinum-etoposide group, respectively. Adverse events occurred in 62% of patients in each arm while adverse events led to death in 5–6% of patients (5% durvalumab plus platinum-etoposide; 6% platinum-etoposide alone). This trial highlighted the utility of durvalumab as a treatment for ES-SCLC. Selected multi-institutional IO trials in SCLC and NSCLC are outlined in Table 2.

However, not all IO regimens have proven successful in the treatment of ES-SCLC. Assessing the addition of pembrolizumab to etoposide and platinum for previously untreated ES-SCLC patients, Keynote-604 showed improved progression-free survival in the combination arm but failed to meet the significant threshold for overall survival [64]. Evaluating nivolumab plus ipilimumab compared to nivolumab alone for maintenance therapy for the treatment of immunotherapy-naïve ES-SCLC patients, CheckMate 451 found the overall survival to be 9.2, 10.4, and 9.6 for nivolumab plus ipilimumab, nivolumab alone, and placebo, respectively [65]. This negative trial suggested the combination of nivolumab plus ipilimumab did not prolong overall survival when compared to placebo as maintenance therapy. Of note, CheckMate 451 assessed the addition of maintenance immunotherapy on overall survival for immunotherapy-naïve, ES-SCLC patients who had already received 3 or 4 cycles of first-line chemotherapy with no disease progression. In comparison, IMpower 133, CASPIAN, and Keynote-604 studied the addition of immunotherapy to first-line chemotherapy for treatment-naïve patients.

In addition to IO and chemotherapy, ES-SCLC can be treated with palliative radiation to symptomatic sites and consolidative thoracic RT to 30 Gy in 10 fractions in patients who respond to chemo-immunotherapy [64]. In a phase III trial prior to the introduction of IO, Slotman et al. found improved OS in patients with residual intrathoracic disease who received thoracic RT, with survival rates at one and two year(s) of 33% (vs. 26%) and 12% (vs. 3%), respectively [65]. This improved 2-year survival with thoracic RT is consistent with prior trials’ delayed survival benefit. However, the discussed above studies establishing the role of ICI in ES-SCLC precluded consolidative thoracic RT. There are ongoing investigations into combining RT with IO, including the RAPTOR Trial, also known as NRG-LU007, that explores the addition of consolidative RT to IO for ES-SCLC. Selected practice-changing trials for IO in SCLC and NSCLC are outlined in Table 2.

**Table 2 biomedicines-11-01642-t002:** Selected practice-changing, multi-institutional clinical trials for IO in SCLC and NSCLC.

Trial Name and Agent	Disease and Stage; Trial Phase; n	Treatment Arms	Primary Objective	Timing of IO in Relation to RT/CHT	Results (Focusing on Primary Objective)	Conclusions
IMpower133 [61] (atezolizumab)	ES-SCLC; III; n = 403	CHT +/− atezolizumab	OS, PFS	Concurrent and adjuvant	Median OS 12.3 mos in the CHT + IO group vs. 10.3 mos in the CHT group. Median PFS 5.2 mos vs. 4.3 mos, respectively.	Increased OS and PFS with CHT + atezolizumab.
CASPIAN [62,63] (durvalumab)	ES-SCLC; III; n = 805	CHT vs. CHT + durvalumab vs. CHT + durvalumab + tremelimumab	OS	Concurrent	Median OS 12.9 mos in the CHT + durvalumab group vs. 10.4 months in the CHT + duvalumab + tremelimumab group vs. 10.5 mos in the CHT group.	Increased OS with CHT + durvalumab.
CheckMate 816 [66] (nivolumab)	Resectable stage IB to IIIA NSCLC; III; n = 358	CHT +/− nivolumab	EFS, pathologic complete response	Concurrent prior to surgery	Median EFS 31.6 mos in the CHT + IO group vs. 20.8 mos in the CHT group. Pathologic complete response 24.0% and 2.2%, respectively.	Increased EFS and pathological complete response with CHT + IO.
PACIFIC [29,30] (durvalumab)	Locally advanced, unresectable NSCLC; III; n = 709	Placebo vs. durvalumab	PFS, OS	Adjuvant	Median PFS 16.9 mos in the IO group vs. 5.6 mos in the placebo group. Median OS 47.5 mos in the IO group vs. 29.1 mos in the placebo group.	Increased PFS and OS with IO.
Keynote-042 [67] (pembrolizumab)	Locally advanced or metastatic NSCLC; III; n = 1274	CHT vs. pembrolizumab	OS in patients with a TPS of ≥50%, ≥20%, or ≥1%	N/A	Median OS by TPS group: - TPS ≥ 50% group-20.0 mos in the IO group vs. 12.2 mos in the CHT group. - TPS ≥ 20% group-17.7 mos in the IO group vs. 13.0 mos in the CHT group. -TPS ≥ 1 group-16.7 mos in the IO group vs. 12.1 mos in the CHT group.	Increased OS with IO in all three TPS groups.
Checkmate 227 [68] (nivolumab, ipilimumab)	Stage IV or recurrent NSCLC; III; n = 1739	CHT vs. CHT + nivolumab vs. CHT + nivolumab + ipilimumab	OS, PFS	N/A	Median OS for patients with PD-L1 expression of: - ≥1%: 17.1 mos in the nivolumab + ipilimumab group vs. 13.9 mo in the CHT group. - <1%: 17.2 mos in the nivolumab + ipilimumab group vs. 12.2 months in the CHT group. Median OS for all patients: 17.1 mos in the nivolumab + ipilimumab vs. 13.9 mos in the CHT group.	Increased OS with nivolumab + ipilimumab.
CheckMate 9LA [69] (nivolumab + ipilimumab)	Stage IV or recurrent NSCLC; III; n = 1150	CHT +/− nivolumab with ipilimumab	OS	Concurrent	Median OS 15.6 mos in the CHT + IO group vs. 10.9 in the CHT group.	Increased OS with CHT + IO.

CHT—chemotherapy. DFS—disease-free survival. EFS—event-free survival. IO—immunotherapy. OS—overall survival. PFS—progression-free survival. TPS—Tumor Proportion Score. RT—radiotherapy.

## 4. NSCLC

### 4.1. Background

NSCLC is the most common type of lung cancer, comprising approximately 85% of all lung cancer cases [70]. It consists of three main subtypes, listed from most to least common: adenocarcinoma, squamous cell carcinoma, and large cell. The risk factors, clinical presentation, and workup of NSCLC are similar to those of SCLC, except brain MR is usually reserved for certain select IB patients and all stage II and above patients [71]. In 2017, approximately 29% of all NSCLC patients in the U.S. are diagnosed with stage I, 8% with stage II, 13% with stage IIIA, 6% with stage IIIB, and 44% with stage IV, with a 5-year survival rate of 68%, 45%, 26%, 17%, and 6%, respectively, at each of the aforementioned stages [72]. The current treatment paradigm for each stage is addressed below. Given many molecular biomarkers, such as *EGFR*, *KRAS*, *ALK*, *ROS1*, *BRAF*, *MET*, and *RET*, with targetable small molecule inhibitors, next generation sequencing is routinely performed in NSCLC. Additional details will be provided regarding the role of IO in NSCLC [73].

### 4.2. Treatment Paradigm for Early Stage NSCLC

In comparison to SCLC, early stage NSCLC are mainly managed with either surgery or RT, with the addition of chemotherapy for select stages. Per the National Comprehensive Cancer Network (NCCN) guidelines, the standard of care for operable patients with negative mediastinal nodes is surgical resection and lymph node dissection or sampling [71]. The standard of care for inoperable patients without nodal disease is definitive RT alone, preferably stereotactic body radiation therapy (SBRT). Exact RT dosing is dependent on the size and location of the lesion, specifically peripheral versus central location. Chemotherapy is usually reserved for select stages IB and IIA and all stages IIB and above. There is an emerging role for IO in the current standard of care, including neoadjuvant chemotherapy with nivolumab for stage IB and II as per CheckMate 816 as discussed below. Ongoing trials are evaluating the integration of ICI for higher risk early state patients, including SWOG S1914, PACIFIC-4, and KEYNOTE-867 that all explore the addition of ICI to SBRT in early stage NSCLC, as reviewed in Table 3.

### 4.3. Treatment Paradigm for Locally Advanced Stage NSCLC

For stage III NSCLC, a combination of surgery, RT, and systemic therapy may be offered to patients with good performance status. For those undergoing surgery, there may be neoadjuvant or adjuvant therapy. Trimodality therapy of neoadjuvant chemotherapy with or without RT of 45–54 Gy in 25 fractions followed by surgery has traditionally been offered in an attempt to downstage the disease. Recently, CheckMate 816 demonstrated that neoadjuvant chemotherapy with nivolumab for treatment-naïve patients with resectable stage IB to IIIA disease allow for increased pathological complete response and longer event-free survival when compared to those receiving neoadjuvant chemotherapy alone [66]. Of note, this trial did not permit neoadjuvant RT, though it did allow RT as an adjuvant treatment. For those undergoing surgery first, a combination of adjuvant chemotherapy, RT, and IO may be offered depending on the presence of disease at the surgical margin. Adjuvant RT is typically offered for positive margins, not for N2 disease, based on recent trial results [81]. The inclusion of IO is reserved for patients with select disease stage with or without certain mutations. For those not receiving surgery, definitive concurrent chemoradiotherapy of 60–70 Gy in 30–35 fractions followed by one year of durvalumab is a category 1 recommendation as per the PACIFIC trial [29].

To date, definitive concurrent chemoradiotherapy followed by consolidative durvalumab is the only recommendation by NCCN for RT with IO in lung cancers, though there are other recommendations for IO with chemotherapy and/or surgery [71]. The PACIFIC trial compared durvalumab versus placebo as consolidative therapy, given every two weeks for up to one year, for stage III NSCLC, immunotherapy-naïve patients who completed definitive chemoradiotherapy without disease progression. A total of 713 patients were randomized. Primary end points included progression-free survival and overall survival, though the interim analysis only reported the first end point, secondary end points, and safety profile [29]. The interim analysis demonstrated that the median progression-free survival is 16.8 months with durvalumab compared to 5.6 months with placebo. Durvalumab also appeared to be superior to placebo in regards to the secondary end points, while the safety profile was comparable between the two arms. The updated 5-year outcomes demonstrated sustained overall survival benefit of 47.5 months with durvalumab versus 29.1 months with placebo and progression-free survival of 16.9 months with durvalumab versus 5.6 months with placebo [30]. Subgroup analyses demonstrated that patients with PD-L1 expression <1% did not enjoy the overall survival and progression-free survival benefits with durvalumab. The supplemental analysis of the PACIFIC trial also suggested a greater benefit in patients who got durvalumab within 14 days of completing CRT, which may be due to greater synergy but may also represent bias toward patients with lower burden of disease or toxicity. The PACIFIC trial remains active with an estimated study completion date in June 2023, with final analysis incoming [82]. Additional clinical trials are exploring the synergistic role of IO and RT in neoadjuvant and adjuvant settings for NSCLC (Table 3). For locally advanced stage NSCLC, several ongoing trials, such as NCT04092283, PACIFIC-8, NCT04929041, and NCT03867175, assess different combinations of chemotherapy, immunotherapy, and RT.

### 4.4. Incorporation of RT and IO for Metastatic Disease

Although durvalumab is part of the treatment paradigm for locally advanced NSCLC, it is not for metastatic NSCLC as per NCT02888743, a phase II trial that failed to demonstrate increased overall response rates for RT with durvalumab plus tremelimumab [83]. For patients with stage IV disease, systemic therapy is typically the first-line treatment and the presence or absence of actionable molecular biomarkers guides the selection of first-line IO. Selected, multi-institutional trials for PD-L1 will be briefly mentioned to demonstrate the evolving standard of care with IO. Initially, multiple trials have demonstrated the efficacy of immunotherapeutic agents alone without surgery, RT, or chemotherapy. For example, treatment-naïve patients with PD-L1 ≥ 1% may be treated with the following IO alone as the first-line therapy: pembrolizumab as per KEYNOTE-042 [67], and nivolumab plus ipilimumab as per CheckMate 227 [68]. Both of these options were compared to chemotherapy, which was the original standard of care, and found to be superior in terms of overall survival. Once the benefits of these immunotherapeutic agents were demonstrated, additional trials were performed to see whether there are additional benefits, including synergistic effects, when combined with other therapies. Several trials explored the addition of IO to the standard of care chemotherapy, such as pembrolizumab to pemetrexed and a platinum-based drug for non-squamous NSCLC, systemic therapy-naïve patients in KEYNOTE-189 [84], pembrolizumab to carboplatin and paclitaxel or nab-paclitaxel for squamous NSCLC, systemic therapy-naïve patients in KEYNOTE-407 [85], nivolumab and ipilimumab to the standard of care chemotherapy based on etiology in CheckMate 9LA for systemic therapy-naïve patients [69], and atezolizumab to carboplatin and nab-paclitaxel in IMpower131 for chemotherapy-naïve and immunotherapy-naïve (barring anti-CTLA-4 therapy) patients [86].

Retrospective studies, including the secondary analysis of KEYNOTE-001 phase I trial, have suggested benefits of IO in patients who have received previous RT [42]. However, the PEMBRO-RT phase II trial, which randomized patients to pembrolizumab alone versus pembrolizumab after receiving RT to a single tumor site, failed to demonstrate a pre-specified meaningful clinical benefit though there is a trend toward improved progression-free survival and overall survival for the IO plus RT arm versus the IO alone arm (6.6 months vs. 1.9 months, 15.9 months vs. 7.6 months, respectively) [32]. Of note, the trial also demonstrated a doubling of the overall response rate from 18% to 36% with the addition of RT to a single tumor site, emphasizing the need for further testing. The recently opened Alliance A082002 trial will randomize patients with PD-L1 negative advanced NSCLC to systemic therapy with or without SBRT (24 Gy in 3 fractions) to a single tumor site with a primary endpoint of improved overall survival [78,79].

RT/IO integrations have also been studied in the context of oligo-metastatic disease states where radiation is added after a good response to systemic therapy in order to provide local control at limited metastatic sites. A phase II single arm trial by Bauml et al. demonstrated a median progression-free survival of 19.1 months (compared with historical control of 6.6 months) for patients with oligometastatic NSCLC with up to four metastatic lesions treated with local therapy to all known sites of disease followed by pembrolizumab 4–12 weeks later [87]. The ongoing NRG LU002 trial was modified to include patients treated with IO agents and randomizes patients with three or fewer sites of oligometastatic disease with disease control to maintenance therapy with or without SBRT to primary disease site [88]. These studies will offer key insights into the role of local therapies include RT for control of metastatic disease and improvement of response to systemic therapies.

## 5. Conclusions

The advances of novel therapeutic approaches and ongoing clinical trials are reshaping the standard of care for SCLC and NSCLC. The critical role of RT is also thus evolving, though the rationale for integrating IO and RT persists and is worthy of continuous exploration. There are multiple ongoing trials assessing the role of RT with IO in lung cancer across stage and histology. Future directions may include assessing the optimal RT dose, fractionation regimen, and timing schedule when integrating IO and RT. While large randomized studies may affect the standard of care, small scientific-focused or preclinical studies can help address these particular questions and elucidate the immune regulatory pathway, thereby further tailoring the large randomized studies.

## Figures and Tables

**Table 1 biomedicines-11-01642-t001:** Selected, ongoing, multi-institutional SCLC trials.

	Trial	Phase	Estimated Enrollment	Stage	Treatment Arms	Primary Objective	Treatment Sequence of IO and RT	Estimated Completion
Adding IO to RT	NRG-LU005 [55]	II/III	n = 506	Limited	CRT +/− atezolizumab	OS	IO given during CRT and continued for 1 yr as maintenance	28 December 2026
	ADRIATIC [56]	III	n = 730	Limited	CRT vs. CRT + durvalumab +/− tremelimumab	PFS, OS	IO given after CRT	5 September 2024
	NCT04624204 [57]	III	n = 672	Limited	CRT vs. CRT + pembrolizumab +/− olaparib	PFS, OS	Pembrolizumab given during CRT and continued for 1 yr as maintenance; olaparib as maintenance	28 October 2027
Adding RT to IO	RAPTOR (NRG-LU007) [58]	II/III	n = 138	Extensive	Atezolizumab +/− RT	PFS, OS	RT given in daily fractions weeks 1–5; IO continued q21 days in the absence of progression or toxicity	30 April 2027

CHT—chemotherapy. CRT—chemoradiotherapy. IO—immunotherapy. OS—overall survival. PFS—progression-free survival. RT—radiotherapy.

**Table 3 biomedicines-11-01642-t003:** Selected, ongoing, multi-institutional NSCLC trials.

	Trial	Phase	Estimated Enrollment	Stage	Treatment Arms	Primary Objective	Treatment Sequence of IO and RT	Estimated Completion
Adding IO to RT	SWOG S1914 [55]	III	n = 480	Early	SBRT +/− atezolizumab	OS	IO given for total of 8 cycles, with SBRT starting in cycle 3	1 May 2028
	PACIFIC-4 [74]	III	n = 733	Early	SBRT vs. SBRT + durvalumab vs. SBRT + osimertinib	PFS	Durvalumab given with SBRT and continued; osimertinib started after SBRT	31 January 2028
	KEYNOTE-867 [75]	III	n = 530	Early	SBRT +/− pembrolizumab	EFS	IO given during and after SBRT	1 July 2026
	NCT04092283 [76]	III	n = 660	Adv	CRT + consolidation durvalumab +/− concurrent durvalumab	OS	IO given prior to, continued during, and after RT vs. after RT only	31 October 2028
	PACIFIC-8 [77]	III	n = 860	Adv	CRT + durvalumab +/− domvanalimab	PFS	IO given monthly for one year after CRT	28 September 2029
Adding RT to IO	Alliance A082002 [78,79]	II/III	n = 427	Adv	IO (+/− CHT) +/− SBRT	PFS, OS	3 fractions SBRT given every other day	31 December 2027
	NCT03867175 [80]	III	n = 112	IV	Pembrolizumab +/− SBRT	PFS	3–10 treatments of SBRT while undergoing 1 year of IO	31 December 2027

Adv—advanced. CHT—chemotherapy. CRT—chemoradiotherapy. IO—immunotherapy. OS—overall survival. PFS—progression-free survival. SBRT—stereotactic body radiotherapy. RT—radiotherapy.

## Data Availability

No new data were created or analyzed in this study. Data sharing is not applicable to this article.

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
