# Peer review of "Radiotherapy and Immunotherapy in Lung Cancer"

_biomedicines, 2023, doi:10.3390/biomedicines11061642_

Round 1

Reviewer 1 Report

This article summarizes published and ongoing clinical studies investigating the combination of radiotherapy and immune checkpoint blockades (ICBs) for NSCLC and SCLC in definitive and metastatic settings. This article may have interest of radiation oncologists. 

The reviewer has the following concerns.

1.     Negative studies for radiotherapy combining ICBs in metastatic NSCLC (e.g. Lancet Oncol 2022;23:279-97) were not mentioned.

2.     Radiation target (single vs. multiple vs. all lesions) and patients’ characteristics (immunotherapy-naïve vs. immunotherapy-resistant) are also important issues for combination of radiotherapy and immunotherapy. It is recommended to discuss these issues.

Reviewer 2 Report

 General comments:

The manuscript should start with a method section. The reader should be advised of many important aspects of how the systematic review has been conducted (the search equation, The explored database, the selection criteria, the evaluation of risk of bias study-by-study, etc.). The PRISMA checklist should be used.

The authors should define precisely in this method section what was the perimeter of their review. The title said: combining immunotherapy and radiotherapy. For the reader, combined therapy means concurrent therapy. Therefore, the PACIFIC study that demonstrated the survival benefit of consolidation Durvalumab in stage III NSCLC responders to CTRT is not a combination of RT and IO insofar as there are used sequentially.

Actually, the intent of the authors is unclear. Did they want to limit the study to combined IO RT in lung cancer (consequently using only concurrent associations) or did they wish to cover more extensively other kind of associations? In the latter case, the authors should consider another title such as immunotherapy and radiotherapy in lung cancer, systematic review of clinical trials testing a synergy.

Specific comments:

Introduction

The authors should consider the globocan publication (doi: 10.3322/caac.21660 Available online at cacancerjournal.com). The most incident cancer type in women is breast cancer and in men, prostate cancer. Worldwide, the breast cancer remained until 2020 the first cause of cancer deaths in women.

The abscopal effect is still debatable and should not appear as the first rationale for combining IOs and RT. Moreover, there are publications on abscopal effect in lung cancer, more recent and published in high IF journals than the references 15-17 (in the current manuscript, the ref 34 [i.e. Lancet Respir Med. 2021 May;9(5)] comes to distantly from abscopal effect citations).

The damage associated molecular patterns and phenotypic of tumor infiltrating cell should be indicated.

There are other attempts to enhance response to IO by adding RT in metastatic lung cancer: adding cyrotherapy (for instance, NCT04793815).

SCLC

In the paragraph reporting the vaccine trials for LS-SCLC, the authors should more accurately indicate the design of the study: these studies only accrued responding LS-SCLC after CRT for adjuvant vaccine therapy.

The authors said regarding the CASPIAN trial « This trial highlighted the utility of durvalumab as a treatment for ES-SCLC while we eagerly await final trial results ». The authors should be advised that the 3-year survival has been published (Goldman JW et al. Lancet Oncol 2021; 22: 51–65) with almost 40-month median follow-up that is largely sufficient in SCLC to consider the study results as the final report.

A third trial comparing CT vs CT and pembroliumab has been published. The combined arm failed to statistically improve the OS.

The CheckMate 451 has a completely different design and goal than did the IMpower 133 and the CASPIAN studies.

The study by Slotman et al (consolidation thoracic RT in ES-SCLC responders to chemotherapy) as been published before the IO area. The authors should delete this study from their review inasmuch as the title is « combining immunotherapy and radiotherapy ». As a point of fact, the all paragraphs on ES-SCLC is debatable in this review because thoracic RT, and even IPC were not allowed in the IO-CT groups for both IMpower 133 and CASPIAN. The author should delete this paragraph.

In table 2 the authors listed their choice regarding practice-changing, multi-institutional clinical trials for IO in SCLC and NSCLC. In a systematic review, it is not possible to make a selection because of the recommendation to be as exhaustive as possible (PRISMA guidelines as above-mentioned). Moreover, the herein review focuses on « combining immunotherapy and radiotherapy ». Only one of the nine trials listed in the table 2 (PACIFIC) combined RT with IO (in a sequential design). The other eight studies, were IO trials in metastatic disease or in surgically resected NSCLC without any intervention of RT in the investigational groups. The authors should delete this section.

NSCLC

The neoadjuvant paradigm is not to reduce the risk of R1 resection; this is a secondary endpoint. The primary endpoint is to down-stage the disease particularly to down-stage nodal status.

The authors should explain why their reported ADAURA, a study testing adjuvant osimertinib versus placebo in EGFR mutant resected NSCLC (IO should not be prescribed in EGFR mutant NSCLC). The ADAURA study does not encompass radiotherapy. The author should delete this study from the text and from the tables. In point of fact, neither IO nor RT were included in the design of the ADAURA study. Therefore, the section should be deleted.

Retrospective studies demonstrating the benefits of IO in patients who have received previous RT, whereas the design consisted of testing the effect of adding IO in metastatic NSCLC is a too weak clue to be cited as a proof of usefulness of RT IO association.

SABR-COMET is out of the scope of the subject of this review insofar as this study did not include IO in the design.

IMpower 010 does not test IO and RT combination. Actually, patient did not received RT which does not improve resected NSCLC (LUNG ART study published a year ago in Lancet oncol by Lepechoux et al). The mention of 010 study should be deleted.

Conclusion

Taking into account the aforementioned specific comments, the reader considers that the conclusion of this review “Lung cancer, specifically SCLC and NSCLC, is the one cancer where there is high quality data for IO and RT across stages and histology “goes far beyond what the current data support. The proofs are weak and the main studies testing RT plus IO combination are still ongoing or not yet analysed. This systematic review comes too early. In addition, most of the study presented did not associate IO and RT in their design and are therefore out of scope taking into account the review title.

Round 2

Reviewer 2 Report

all queries have been taken into account.